# Prospects of and Barriers to the Development of Epitope-Based Vaccines against Human Metapneumovirus

**DOI:** 10.3390/pathogens9060481

**Published:** 2020-06-18

**Authors:** Ekaterina Stepanova, Victoria Matyushenko, Larisa Rudenko, Irina Isakova-Sivak

**Affiliations:** Department of Virology, Institute of Experimental Medicine, Saint Petersburg 197376, Russia; fedorova.iem@gmail.com (E.S.); matyshenko@iemspb.ru (V.M.); rudenko.lg@iemspb.ru (L.R.)

**Keywords:** human metapneumovirus, HMPV vaccine, impaired immune responses, epitope-based vaccines, viral vectors, conserved HMPV epitopes, memory T cells

## Abstract

Human metapneumovirus (HMPV) is a major cause of respiratory illnesses in children, the elderly and immunocompromised patients. Although this pathogen was only discovered in 2001, an enormous amount of research has been conducted in order to develop safe and effective vaccines to prevent people from contracting the disease. In this review, we summarize current knowledge about the most promising experimental B- and T-cell epitopes of human metapneumovirus for the rational design of HMPV vaccines using vector delivery systems, paying special attention to the conservation of these epitopes among different lineages/genotypes of HMPV. The prospects of the successful development of an epitope-based HMPV vaccine are discussed in the context of recent findings regarding HMPV’s ability to modulate host immunity. In particular, we discuss the lack of data on experimental human CD4 T-cell epitopes for HMPV despite the role of CD4 lymphocytes in both the induction of higher neutralizing antibody titers and the establishment of CD8 memory T-cell responses. We conclude that current research should be focused on searching for human CD4 T-cell epitopes of HMPV that can help us to design a safe and cross-protective epitope-based HMPV vaccine.

## 1. Introduction

Human metapneumovirus (HMPV) was first isolated in the Netherlands in 2001 [1], but has been circulating unrecognized for decades due to its clinical manifestation being similar to that of other respiratory viruses and difficulties with its propagation in cell cultures [2]. HMPV, along with influenza and respiratory syncytial virus (RSV), causes a significant worldwide socio-economic burden and mostly affects infants and children younger than 5 years of age [2,3]. HMPV was shown to have an even higher impact than RSV on the development of hypoxic respiratory illnesses in asthmatic children [4]. Globally, HMPV accounts for 6–40% of all hospitalized and outpatient children with an acute respiratory illness [2]. Overall, it is believed that most children experience an HMPV infection by the age of 5 [1,5,6]. Despite this seroprevalence, HMPV infections reoccur throughout the adult life, though in healthy adults the infection is usually asymptomatic or mild if present [7]. Other high-risk groups for HMPV complications include elderly people and adult patients with underlying medical conditions [8,9,10]. HMPV’s seasonality is similar to that of other respiratory viruses; it peaks during the winter and spring months in temperate regions and circulates throughout the year in tropical countries [3].

Metapneumovirus is an enveloped RNA virus that belongs to the *Pneumoviridae* family [11]. The virus has a single-stranded antisense RNA genome with eight genes encoding nine viral proteins. Each viral particle is enveloped in a lipid bilayer that embeds a fusion protein, an attachment glycoprotein (G), and a short hydrophobic (SH) protein. Inside the capsid, which is formed by matrix protein (M), is located a nucleocapsid that includes viral RNA in a complex with a nucleoprotein (N), a polymerase protein (L), and a phosphoprotein (P). HMPV is divided into the genetic lineages A and B on the basis of the N, F, G, and L gene sequences. Each line includes two genotypes (A1, A2, B1, and B2) and several sub-genotypes (A2a, A2b, A2c, B2a, and B2b) [3]. Studies on animal models suggest that immunization against HMPV of one genetic lineage produces cross-neutralizing antibodies and protects against infection with HMPV of another lineage [12,13].

Unlike influenza, no vaccine or specific prophylaxis for HMPV has been approved for human use, despite the tremendous efforts of researchers around the world [14]. A vaccine for HMPV is essential to alleviating the burden of disease, especially given the unique ability of the virus to modulate the host’s immunity, resulting, after a natural HMPV infection, in the development of a weak adaptive immunity that leads to recurrent infections [7,15]. Classical approaches to the development of HMPV vaccines are not suitable for several reasons. For example, administration of an alum-adjuvanted, formalin-inactivated vaccine against the closely related RS virus to infants and toddlers in the 1960s led to enhanced pulmonary disease (EPD) in vaccinated children who were subsequently infected with natural RSV; two cases were fatal [16]. Similar outcomes for a formalin-inactivated HMPV vaccine were obtained using different animal models, such as cotton rats [17] and macaques [18], indicating that this vaccine should not be transferred to clinical evaluation. Another classical approach to the development of vaccines against respiratory viruses involves attenuation of the pathogen by culturing it under specific conditions. Such live attenuated vaccines are administered intranasally in order to establish antiviral immunity at the port of viral entry. One such candidate was developed by serial passaging of HMPV in Vero-83 cells at gradually decreasing temperatures, yielding a temperature-sensitive virus whose replication was restricted to the upper respiratory tract of Syrian golden hamsters [19]. Although this vaccine induced cross-protective immunity and protected animals from HMPV pulmonary replication, the presence of all molecular determinants that HMPV uses to modulate host immunity indicated that this vaccine would provide only low potential immunogenicity to humans.

More advanced strategies for generating recombinant live attenuated HMPV vaccines have been explored in recent years, yielding promising results in animal models [7,20,21]. The most advanced recombinant live attenuated HMPV vaccine was tested in a phase I clinical trial in adults and children. The results showed that the vaccine was overattenuated in HMPV-seronegative children, which led to the trial’s termination [22]. A number of recombinant protein or virus-like particle (VLP)-based HMPV vaccine candidates have also been evaluated in pre-clinical studies [7,20]. Most of the protein-based vaccines expressed the full-length fusion protein (F), which is the major antigenic determinant of the virus. These vaccines were capable of inducing HMPV-neutralizing antibodies in animal models with high protective potential [23,24]. However, there is evidence that the number of such antibodies can rapidly wane over time, which represents a major limitation of this kind of vaccine [25]. Furthermore, it is known that HMPV can spread directly from cell to cell in the presence of neutralizing antibodies, without requiring the attachment factor, through the use of the actin cytoskeleton [26,27]. These findings suggest that a successful HMPV vaccine should also induce protective T-cell-mediated immunity—in particular, cytotoxic T lymphocytes (CTLs), which can kill virus-infected cells. Replicating viral vectors that are administered intranasally represent a promising strategy for the delivery of HMPV antigens directly to the respiratory tract and the induction of protective immunity at the site of infection. To date, such viral vectors as recombinant human parainfluenza virus type 1 (rHPIV1) [28], Sendai virus (a murine PIV type 1 virus that is closely related to HPIV1) [29], and bovine/human chimeric parainfluenza virus type 3 [30] have been successfully used to design viral-vector-based HMPV vaccines. These viruses, when administered intranasally, can induce secretory IgA antibodies, as well as durable B- and T-cell immunity, at both systemic and local levels [31,32,33], thus increasing their potential to provide long-lasting protection against natural HMPV infections.

Attenuated influenza viruses represent another attractive platform for the development of HMPV vaccines due to the ability of influenza viruses to induce long-lasting local and systemic immunity. Influenza vaccination campaigns are routinely implemented in many countries, and live attenuated influenza vaccines (LAIVs) have been used in clinical practice for decades without safety concerns [34,35]. In addition, attenuated influenza viruses have been shown to establish lung-localized memory T-cell (T_RM_) responses, and these T_RM_s can recognize evasive pathogens and promptly respond to the infection by rapidly proliferating effector T cells [36]. However, unlike other viral vectors, the capacity of influenza virus genomes to insert foreign antigens is limited. In this review, we summarize current knowledge about the most promising experimental B- and T-cell epitopes of human metapneumovirus for the rational design of HMPV vaccines using vectors with a limited genome capacity, paying special attention to the conservation of these epitopes among different lineages/genotypes/sub-genotypes of HMPV. The prospects of the successful development of an epitope-based HMPV vaccine are discussed in the context of recent findings regarding HMPV’s ability to modulate host immunity.

## 2. Peculiarities of the Development of Immune Responses to HMPV Infection

In order to induce protective immune responses against HMPV without the simultaneous development of enhanced pulmonary disease, it is critical to take into account the specific mechanisms this virus uses to modulate host immunity. In humans, higher levels of IgG and HMPV-neutralizing antibodies were associated with protection against HMPV during the winter seasons [37]. In animal models, primary HMPV infection induced weak and aberrant adaptive immune responses; however, a passive transfer of hyperimmune HMPV-specific mouse sera significantly reduced HMPV titers in mouse lungs [38]. These data suggest that high levels of virus-specific antibodies can contribute to protection against HMPV infection, but humoral immunity alone cannot always control an HMPV infection, and the virus can persist in mice even in the presence of neutralizing antibodies [39]. After natural infection with HMPV, protective immunity was shown to be transient [40]. Furthermore, attempts to induce high levels of neutralizing titers using inactivated HMPV vaccines resulted in the development of an enhanced pulmonary pathology in vaccinated animals after challenge; these results were reaffirmed in mice [41], cotton rats [17], and macaques [18]. Despite high numbers of neutralizing antibodies, a low viral load was detected in lungs of immunized animals, and immunopathology in the lungs was found to be due to aberrant T-cell immunity and an increased Th-2 response [41,42].

T-cell immunity to HMPV infection plays an important role in virus control, but it can also contribute to pathology. In experiments on mice with T cell depletion, both CD4 and CD8 T cells were required for efficient viral clearance during primary HMPV infection; however, both subsets played a role in the enhancement of disease manifestation. Depletion of one of the subsets showed that both CD4 and CD8 T cells were engaged in the development of a lung pathology, with a greater involvement of CD4 T cells. Although CD4-depleted mice had impaired generation of neutralizing antibodies, they were fully protected against HMPV replication and clinical disease, indicating that the presence of the CD8 subset alone can be sufficiently protective [43]. In line with this, a passive transfer of HMPV-epitope-specific CD8 T cells to RAG-1-deficient mice (mice lacking mature B and T lymphocytes) reduced the HMPV titer in lungs by ∼30 times four days after challenge, further indicating the importance of CD8 immunity in HMPV clearance [44].

It is believed that a successful vaccine against respiratory viruses should induce memory CD8 T-cell responses. Upon re-encountering a pathogen, these memory T cells can quickly reactivate their effector functions and proliferate, contributing to protection [45]. One of the reasons for recurrent HMPV infections is that this virus induces an impaired CD8 CTL response and does not generate a long-lasting immunological memory. The mechanism of CD8 response suppression, mediated through programmed death-1 (PD-1) inhibitory receptor/ligand interaction, during HMPV infection was studied by Erickson and colleagues [46,47,48,49]. This mechanism of the downregulation of the CD8 T-cell response through the upregulation of PD-1 was found to be similar to the T-cell functional impairment described for chronic infections [50]. Importantly, this functional impairment of CD8 T cells by natural infection with HMPV was noted only for CD8 T cells in the respiratory tract, whereas virus-specific CD8 T cells in lymphoid organs remained functional [46]. In experiments on animal models, blockade of PD-1 prevented the functional impairment of CD8 effector cells in lungs during reinfection with HMPV, which provided enhanced virus clearance [46]. The impairment of the effector CTL response occurs regardless of the origin of memory CD8 T cells in the case of both infection-induced CD8 T cells and peptide-vaccine-induced CD8 T cells. This provides evidence that, in the case of HMPV, the development of effective vaccines requires a special approach [48,49].

During natural infection, HMPV employs different strategies to circumvent the host’s interferon response to the pathogen, thus resulting in impaired innate and adaptive immune responses [51]. HMPV modulates host immunity by inhibiting the activation of cytokine signaling; this mechanism depends on the cell type, and there is some variation in virus strains. In airway epithelial cells, the virus blocks the RIG-I/MAVS signaling system with viral G proteins, which interact with RIG-I [52,53]. In the B1 strain, the same blockade of the pathway was shown to occur with the P protein [54]. The M2-2 protein interacts with MAVS to block this pathway on the next level [55]. In addition, M2-2-dependent interaction with MAVS results in the inhibition of NF-κB transcription, which, on the other side, is controlled by the SH protein [56].

HMPV’s interference with the functions of dendritic cells (DCs), which aims to subvert the activation of both innate and adaptive immune responses, has been studied in detail in a mouse model [15,57]. In DCs, IFN signaling is impaired by virus proteins. In monocyte-derived DCs, the G protein affects TLR-4, inhibiting chemokine and type I IFN expression [58]. The M2-2 protein inhibits α-IFN expression in plasmacytoid dendritic cells through inhibition of MyD88-dependent phosphorylation of IRF-7 [59]. The SH protein was also shown to inhibit TLR7/MyD88 [60]. The involvement of the MDA5 gene in HMPV-induced IFN regulation has also been described [52]. The SH and G proteins were shown to impair the virus’s ability to be internalized by monocyte-derived dendritic cells, thus reducing their ability to activate CD4 T lymphocytes [61]. A yet-to-be-described soluble factor secreted by HMPV-infected DCs is able to downregulate the activation of CD4 T cells [62]. A low capacity of conventional dendritic cells to present antigens to CD4 cells was also confirmed in mice [63].

The main cause of enhanced disease manifestation during HMPV infection is overactivation of Th2 cells and immune-mediated pathologies in airways. As mentioned above, depletion of CD4 was found to significantly decrease the number of lung pathologies in mice [43]. At the onset of the disease, Th1/Th2-mediated immunity pathways are activated simultaneously [64,65]. The thymic stromal lymphopoietin (TSLP) pathway’s activation by HMPV has been shown to play a role in the initiation of an HMPV-induced immunopathology at the onset of the disease through skewing of CD4 differentiation to IL-10 (Treg) and IL-13 (Th2) production [64]. The Treg CD4 subset plays a special role; early Treg depletion was shown to lead to Th2 polarization of the immune response, with enhanced lung pathology and delayed virus clearance. During the priming phase of infection, depletion of Tregs resulted in impaired migration of dendritic cells and CD8 lymphocytes to lymph nodes and lungs. Nevertheless, a complete absence of Tregs led to a reduced virus titer and an enhanced CD8 response. Late Treg depletion (2 days after inoculation with the virus) enhanced CD8 T-cell activity and improved virus clearance [66].

Currently, the mechanisms by which HMPV modulates host immunity are being intensively studied; however, there remain many unsolved problems. The main problem that needs to be solved in order for a vaccine to successfully be developed is that of correctly priming the host for proper immune responses after natural infection with HMPV (i.e., the generation of an immunological memory). The information that we currently have suggests that CTLs, in collaboration with neutralizing antibodies, play a primary role in anti-HMPV protective immunity. The goal of a successful vaccine design is the proper stimulation of CD4 T cells after immunization. This stimulation is needed in order to increase the production of antibodies (with T follicular helpers) and to form a pool of CD8 memory T cells for rapid and effective secondary CTL responses to natural infection with HMPV. Therefore, studying the CD4 branch of HMPV immunity is the key requirement for the control of pathological processes and the development of prevention strategies.

## 3. Overview of Current Approaches to the Development of HMPV Vaccine

As was mentioned earlier, classical approaches to the development of HMPV vaccines were shown to be ineffective. Current strategies for rational HMPV vaccine design include the use of recombinant viral proteins (perhaps in a VLP formulation), recombinant live attenuated HMPV vaccines, and viral-vectored constructs.

### 3.1. Recombinant Protein and VLP-Based Vaccines

A non-replicating VLP-based vaccine, consisting of an M protein and a fusion-competent F protein, was developed by Cox et al. [24]. C57BL/6 mice immunized with VLPs supplemented with adjuvants showed anti-F antibody responses and a cellular immune response to HMPV. The adjuvants had antagonistic effects on the CTL immune response. Later, it was shown that challenge of VLP-immunized animals with HMPV led to a PD-1-mediated impairment in the CD8 response, similar to the mechanism described for HMPV reinfection [67]. In Wen et al.’s study, VLP-based immunization protected mice against challenge with the homologous A2 and heterologous B2 HMPV strains [67]. In another study by Levy et al., a VLP-based vaccine was developed on the basis of a retroviral vector that displayed the surface glycoproteins F and G. It was demonstrated to provide protection after challenge with HMPV in an animal model [23].

### 3.2. Live Attenuated HMPV Vaccines

Several live attenuated vaccines against HMPV are currently under development [14,20,68]. NIAID scientists have developed two attenuated recombinant MPV viruses, on the backbone of a human MPV strain, in which the phosphoprotein (rHMPV-Pa) or nucleoprotein (rHMPV-Na) genes are substituted by the same genes from the avian MPV subtype C [69]. rHMPV-Pa yielded good results in preclinical research, but in clinical trials this live attenuated virus was shown to be overattenuated in hMPV-seronegative children, which form the main target group for vaccination (ClinicalTrials.gov identifier: NCT01255410) [22]. The rHMPV-Na candidate has yet to be tested in humans.

Another group of live attenuated HMPV vaccine candidates consists of genetically modified HMPV strains that lack one or two non-essential viral proteins known to negatively modulate innate and adaptive immune responses, such as G, SH, and M2-2 (ΔSH; ΔG; and ΔM2-2, respectively). Experiments have shown that it is possible to delete the G, SH, and M2-2 genes while retaining virus infectivity. Such engineered viruses were attenuated in non-human primates and protected animals against reinfection [70]. SH deletion did not result in virus attenuation in a hamster model [71]. G and M2-2 deletion led to virus attenuation in hamsters and non-human primates [70,72]. Two groups reported that M2-2 deletion in HMPV resulted in virus attenuation in hamsters, and all immunized animals developed high levels of virus-neutralizing antibodies and were protected against reinfection after challenge with HMPV [72,73].

More recent approaches to the rational design of recombinant live attenuated HMPV vaccines include the removal of the N-linked carbohydrate at the 172nd amino acid of the F protein of an HMPV strain [74] and the inhibition of the methyltransferase domain of viral RNA polymerase [75]. These vaccines were successfully tested in animal models and shown to be attenuated, immunogenic and provide protection against reinfection with HMPV.

Overall, the use of live attenuated vaccines against HMPV infection can be considered to be a promising strategy for vaccine design since these vaccines are administered intranasally and are capable of stimulating both humoral and cell-mediated immunity. Importantly, these vaccines will provide maximal HLA allele population coverage since the majority of immunogenic T-cell epitopes are present in the attenuated HMPV viruses. On the other hand, due to the complicated nature of the interactions between HMPV and the host’s immune system, these engineered viruses might include yet-to-be-described factors that contribute to modulating the host’s immunity. Therefore, any further clinical development of these candidates should include a broad assessment of immunogenicity and the duration of the protective response. Special attention should be paid to experiments involving specific models to estimate the respiratory pathology after challenge with HMPV.

### 3.3. Vectored HMPV Vaccines

The use of other respiratory viruses as vector delivery systems seems to be the most promising strategy for the rational design of HMPV vaccines because these vectors can effectively deliver HMPV antigens to the right cell compartments, resulting in the proper presentation of antigens to the immune system and the development of balanced antibody and cell-mediated immune responses [76]. Viral-vectored vaccines allow us to include into a recombinant virus only the desirable immunogenic fragments of a pathogen and exclude factors known to contribute to pathogenicity and modulation of the host’s immunity. The key issue is to select the delivery system that will provide optimal stimulation of the immune response. Currently, several viral-vectored vaccines against HMPV infection are under development (reviewed in [20]). The majority of viral-vectored vaccines use HMPV’s fusion protein as a target antigen because of its ability to induce robust HMPV-neutralizing antibodies and confer cell-mediated immunity. To date, such respiratory viruses as recombinant bovine/human PIV3 (bovine PIV backbone genes and human PIV3 F and HN surface proteins), human PIV1, and Sendai virus (a murine PIV1 virus) have been explored as potential vector systems expressing HMPV’s F protein, yielding promising results in animal models [20].

Besides heterologous respiratory viruses, other vector delivery platforms have shown promise for inducing protective immunity against HMPV. Virus-replicon particles of the Venezuelan equine encephalitis virus expressing HMPV’s F, but not G, protein were shown to be immunogenic and protective in mice, cotton rats, and African green monkeys after intranasal (mice and cotton rats) or intradermal (monkeys) immunization [77,78]. Bacterial vectors have also been used in rational HMPV vaccine design. For example, the P-recombinant Bacillus Calmette–Guérin (BCG) vaccine induced the expression of helper and cytotoxic memory T cells that promoted HMPV clearance from mouse lungs without the development of pulmonary disease [79,80].

While the majority of viral-vectored HMPV vaccines that have been developed to date express one HMPV protein, other viral proteins can also contain important epitopes for the induction of balanced B- and T-cell immunity with memory potential. The selection of those epitopes, which are distributed among all HMPV proteins and conserved between different HMPV genotypes, may open up perspectives on the development of broadly protective HMPV vaccines using different vector delivery platforms. Importantly, the use of target epitopes, rather than whole genes, to design vaccines is crucial to vectors with a limited genome capacity to insert foreign antigens, such as attenuated influenza viruses [34,81]. Like HMPV, influenza virus is a respiratory pathogen and can induce a local immunity barrier in the airways, making it a promising platform for the delivery of HMPV epitopes to the site of infection. In addition, attenuated influenza viruses are known to be effective inducers of cross-reactive T-cell responses [82,83]. Therefore, current research should be focused on searching for the most promising HMPV epitopes for the rational design of epitope-based HMPV vaccines for humans.

## 4. Prospects of and Barriers to the Development of Epitope-Based HMPV Vaccines for Humans

It is well accepted that epitope-specific memory CD8 T cells are capable of providing protection against reinfection with respiratory viruses [45]. However, to induce protective memory CD8 T cells, help from CD4 T cells is required. A recent study with influenza viruses demonstrated that help from CD4 T cells at the time of priming was required in order to program memory CD8 T cells to engage with the metabolic biological pathways essential to effective recall responses [84]. Therefore, the key issue for the development of T-cell-based vaccines is the choice of immunodominant CD8 and CD4 epitopes of a target pathogen. The incorporation of a single immunodominant epitope of the SARS coronavirus into a vectored vaccine provided protection against lethal infection mediated by tissue-resident memory T cells [85]. In our studies, previously described immunodominant T-cell epitopes of RSV or human adenoviruses incorporated into a live attenuated influenza virus vector induced in mouse models functional T-cell memory responses to both influenza and the pathogen of interest [86,87]. A similar strategy can be applied in the development of an HMPV vaccine after the most promising immunogenic HMPV epitopes have been selected.

### 4.1. Epitope-Based HMPV Vaccines in Animal Models

Several CTL-targeted HMPV vaccines have been studied in animal models. The HBs-Ag vectored vaccine, which expresses a multiepitope cassette that includes an HMPV epitope, was shown to be immunogenic in mice and a CTL response after immunization was confirmed as IFNγ was detected after stimulation with a peptide [88]. In a study by Herd and colleagues, immunization of BALB/c, C57BL/6, and HLA-A2 transgenic mice with CD8-directed peptide vaccines induced an increase in CTL responses after challenge with HMPV and reduced HMPV virus titers in the lungs. In addition, the authors found increased levels of IFNγ and IL-12 cytokines after challenge with HMPV, which allowed them to speculate about CD4 lymphocytes’ preference for the Th1-differentiation pathway [89]. In another study, Li et al. predicted several CD4 and CD8 T-cell epitopes for BALB/c mice using bioinformatics tools [90]. A multi-epitope peptide vaccine induced CTL and balanced Th1/Th2 immune responses in mice; however, no challenge with HMPV was performed to prove that this immunity was indeed protective [90].

Taken together, these studies yield important conclusions regarding the mechanisms of epitope-based vaccines; however, T-cell epitopes have been proven to be immunodominant for non-transgenic mice and thus cannot be used in human vaccines. Only epitopes that are restricted to human MHC alleles should be considered when designing vaccines for human use.

### 4.2. Overview of Human HMPV Immunogenic Epitopes for the Design of Epitope-Based HMPV Vaccines

Compared with RSV, the basis for HMPV epitope immunity is less studied: 773 RSV epitopes have been deposited at the IEDB, whereas only 138 HMPV epitopes have been described in 14 studies. Thirty-seven of these HMPV epitopes were confirmed in T-cell assays and are class I epitopes (Appendix A). One hundred and four epitopes were shown to bind to recombinant human MHC molecules of the HLA-A*01:01, HLA-A*02:01, and HLA-B*07:02 alleles [91]. Eight epitopes are spatial B-cell epitopes in the F0 protein [92,93,94]. CD8 T-cell epitopes have been studied by several groups. Hastings et al. studied HLA-A*02:01-restricted human epitopes in HLA-A2 transgenic mice using adjuvanted peptide immunization; two peptides were also studied in human PBMCs. The M39 peptide was demonstrated to have protective potential in a mouse challenge model [95]. Erickson et al. mapped two CD8 epitopes using HLA-B7 transgenic mice, which can only recognize human HLA-B*0702 CD8 T-cell epitopes [46]. Herd et al. studied human PBMC response and described nine CTL epitopes for common HLA supertypes [96].

#### 4.2.1. B-Cell Epitopes

All HMPV B-cell epitopes identified to date were found in the fusion glycoprotein, which is the main target for neutralizing antibodies in the HMPV virion. Attachment protein G has been shown to be more variable than attachment protein F and, in experimental studies, did not generate protective immunity [12,28,97]. Eight neutralizing B-cell epitopes have been deposited at the Immune Epitope Database (Table 1). The majority of the epitopes were described by Ulbrandt et al. in an escape-mutant study. Epitope groups 2–6 in the HMPV F protein were distinguished on the basis of epitope location [94].

Figure 1 illustrates the localization of all B-cell epitopes that have been deposited at the IEDB. These epitopes are mapped onto the 3D structure of HMPV’s F protein. The epitopes described in [94] are mapped as single residues, reflecting changes found in escape mutants. The structural epitope 174346, which is described in [92,93], is formed by amino acid residues located along the entire length of the sequence. This is the epitope for neutralizing mAb DS7; however, this epitope exists only in a mature protein structure and is irrelevant to the development of epitope-based vaccines. The same situation exists for epitopes for other cross-reactive anti-RSV mAbs that have been shown to neutralize HMPV (MPE8, 25P13) [98]. In terms of epitope-based vaccines, epitope groups 4–6 (described in [94]) are of special interest. Epitopes in group 4 (epitopes 101454, 101455, and 101458) have a similar structure to that of RSV antigenic site II, the rigid helix–turn–helix structure, which is a target for the therapeutic anti-RSV mAbs palivizumab and motavizumab [93,94]. Epitope groups 5 and 6 (epitopes 101452 and 101454) correspond to the antigenic sites IV–VI of RSV’s F protein. In a recent study, the neutralization of HMPV with the anti-RSV mAb 101F was described. This antibody binds to RSV F 427–237 amino acid residues, which correspond to amino acids 395–405 in HMPV F protein [99]. Since these two fragments’ conformation is preserved during the fusion process [93,99], and they involve residues from a limited number of sequence regions, they can be considered when designing epitope-based vaccines. The location of these two fragments in the structure of the HMPV F protein is mapped in Figure 2.

Six more linear B-cell epitopes within HMPV F protein were predicted using bioinformatics tools [90]. Mice immunized with a multi-epitope recombinant protein vaccine elicited antibodies with virus-neutralizing activity. Mapping of the predicted B-cell epitopes showed that peptides 311–333 and 350–369 partially overlap with the DS7 and MPE8 mAbs epitopes reported by Wen et al. [92,100] (Figure 3). It should be noted that the capacity of individual predicted peptides to neutralize HMPV was not determined in Li et al.’s study [90].

#### 4.2.2. T-Cell Epitopes

The development of a successful T-cell-based vaccine for humans has to be based on experimentally confirmed information regarding natural T-cell epitope processing, immunodominant capacities, the ability to induce memory T cells, and HLA allele coverage. In addition, the selected epitopes should be conserved among different HMPV genotypes to generate cross-protective immunity. In experimental studies, natural processing for 24 HMPV epitopes was confirmed. Sixteen were restricted by the MHC-I molecule of the HLA-A*02:01 allele, five were restricted by the HLA-B*07 allele, one was restricted by the HLA-B*11 allele, and two were found in PBMCs of non-typed donors (Appendix A). Some of these epitopes are located within variable protein regions, indicating that they have limited cross-protective potential. A large-scale study with proteome scanning and a survey of the bioinformatically predicted ability of peptides to bind the MHC of three conventional alleles (HLA-A*01:01, HLAA*02:01, and HLA-B*07:02) was performed by Rock et al. [91]. This material provides us with an opportunity to design epitope-based vaccines on the basis of experimentally confirmed information; however, for all epitopes, natural processing has to be confirmed. The second point to consider is the allele coverage of epitope-based vaccines for the human population, as the allele frequency varies dramatically across different geographic regions [102]. The alleles mentioned above are commonly found in the human population with some geographical and ethnic variability. The average frequency of the HLA-A*02:01 allele lies between 0.05 (in distinct Asian population samples) and 0.5 (in South American population samples). This means that, on average, 10–70% of people have this phenotype, depending on the population. The HLA-A*01:01 allele is less common, with 5–35% of the population having this phenotype. HLA-A*11 has a phenotype frequency of 20–25% in non-Asian populations and a much higher frequency (20–60%, up to 89%) in Asia. HLA-B*07:02, in contrast, is more frequently found in European populations (a phenotype frequency of 20–25%) than in Asia (1–10%). These data on average allele frequencies are from the http://www.allelefrequencies.net database.

To narrow down the list of promising CD8 T-cell epitopes based on their cross-protective potential, we performed an alignment of full-genome sequences of 157 HMPV strains of all genotypes deposited at the Virus Pathogen Database (ViPR, viprbrc.org) and mapped all experimentally established HMPV T-cell epitopes deposited at the IEDB to these alignments. Many of the described T-cell epitopes corresponded to highly conserved viral protein regions, indicating their broad reactivity (Figure 4, Table 2). Nevertheless, among the CD8 epitopes with confirmed natural processing, only nine fell within conserved regions. In the F protein, two overlapping epitopes, 547003 and 69387, are in region 153–165, and both are HLA-A*02:01-restricted. Epitope 69387 was studied in human PBMCs. The third epitope, 33979 (F429–438), is of an unknown class I HLA isotype. Two epitopes are in the N protein: 159112 (N39–47) is HLA-A*02:01-restricted, and 60092 (N307–315) is HLA-B*07:02-restricted. Four epitopes were established in the M protein: 539268 (M39–47) is HLA-A*02:01-restricted, 28126 (M12–20) and 158691 (M195–203) are HLA-B*07-restricted, and 25388 (M194–203) was established in PBMCs of an untyped donor, but is in fact the same peptide as 158691; thus, it probably interacts with HLA-B*07:02.

## 5. Conclusions

Despite the relatively short history of HMPV research, much progress has been made in the development and evaluation of anti-HMPV vaccines. Nevertheless, no vaccines have been licensed to date, although several candidates have yielded promising results in animal models. Overall, there are many potential targets for the rational design of epitope-based HMPV vaccines, preferably using vectored platforms. Viral vectors are a promising platform for the development of vaccines against other viruses due to effective immunogen delivery and stimulation of the appropriate components of the immune system specific for intracellular pathogens. Some of the experimentally evaluated neutralizing epitopes and CD8 T-cell epitopes have the potential to provide broad reactivity against various HMPV genotypes. However, more research needs to be done in order to find an optimal way to induce good-quality memory responses to HMPV. In particular, there is a lack of data on experimental human CD4 T-cell epitopes for HMPV, whereas animal studies suggest that proper induction of CD4 responses can help to generate memory immune responses [66,103]. On the other hand, thorough studies of CD4 responses to HMPV infection are required in both humans and animal models. Immunological studies indicate that the most severe cases of HMPV infection are associated with immunopathological mechanisms that may involve CD4 T cells. In any case, the role of CD4 lymphocytes is difficult to underestimate; their participation is necessary for both the induction of higher neutralizing antibody titers (follicular helper T cells) and the enhancement of the CTL response (Th1). In the presence of the studied CD4 epitopes for humans and susceptible animal models, the development of vaccines will allow us to determine the most promising strategies for engineering vaccines with the desired target. A heterologous vector with mucosal administration may effectively deliver HMPV epitopes to the respiratory tract, providing a mucosal antibody response to contain the pathogen whenever possible and memory T-cell responses to control the infection. Finally, an in-vivo PBMC-based system, similar to one developed recently for the assessment of T-cell-based universal influenza vaccines, could be established to evaluate potential vaccine candidates [104]. Since the vast majority of humans experience HMPV infection by the age of 5, there should be some level of immunological memory of this infection that could be activated by epitope-based vaccines. T cells with various memory phenotypes could be assessed by PBMC-based tests using HLA-typed human donor blood. To date, the most promising approaches to the development of HMPV vaccines look like the use of live antigen delivery systems (vectors or live attenuated viruses) with an accurate immunogen design to ensure effective T-cell memory formation. Additional studies of immunodominant CD8 and CD4 epitopes in HMPV proteins are needed; the role of CD4 T-cell subpopulations has to be estimated in detail. Several methods have been described to evaluate HMPV T-cell memory in humans, as well as the efficacy of T-cell vaccines in experimental animal models. These approaches can help develop effective epitope-based vector vaccines for the prevention of HMPV.

## Figures and Tables

**Figure 1 pathogens-09-00481-f001:**
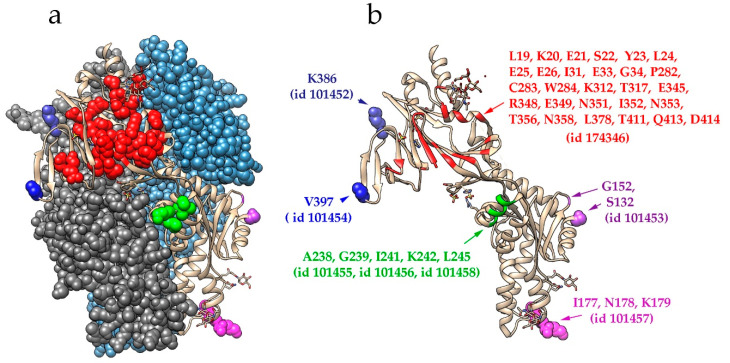
Neutralizing B-cell epitopes mapped onto the structure of HMPV F glycoprotein structure (prefusion state) (pdb ID 5wb0). Individual amino acid residues are indicated. The IEDB epitope ID is specified in brackets. (**a**) F glycoprotein trimer (prefusion state). (**b**) F glycoprotein monomer structure (prefusion state). The figure was prepared using Chimera 1.14 software [101].

**Figure 2 pathogens-09-00481-f002:**
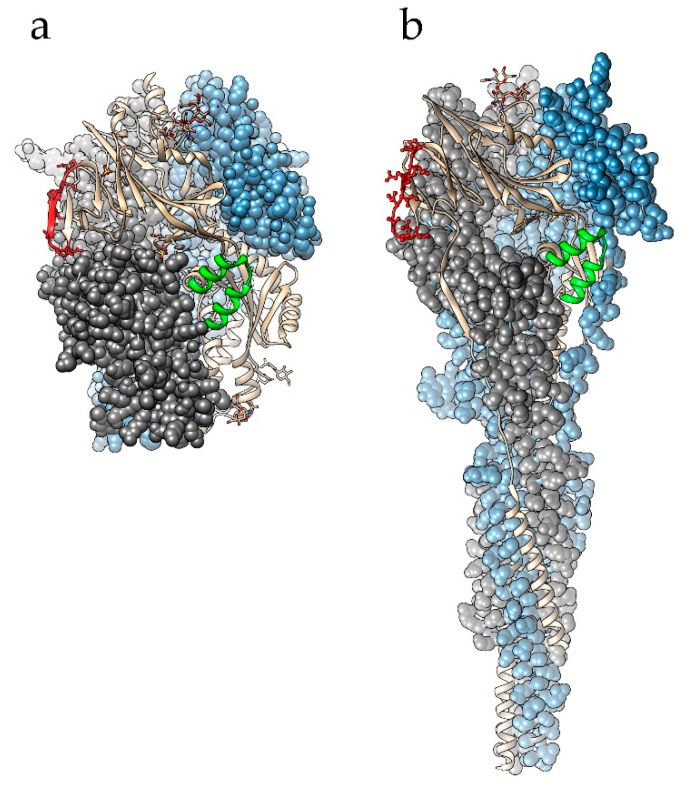
Localization of fragments 395–405 and 224–247 in the structure of HMPV F protein. Fragment 395–405, the predicted site of interaction with 101F mAb [99], is colored in red. Fragment 224–247, the predicted site of interaction with Palivizumab, is colored in green [93,94]. (**a**) Prefusion state, pdb ID 5wb0; (**b**) postfusion state, pdbID 5l1x. The figure was prepared using Chimera 1.14 software [101].

**Figure 3 pathogens-09-00481-f003:**
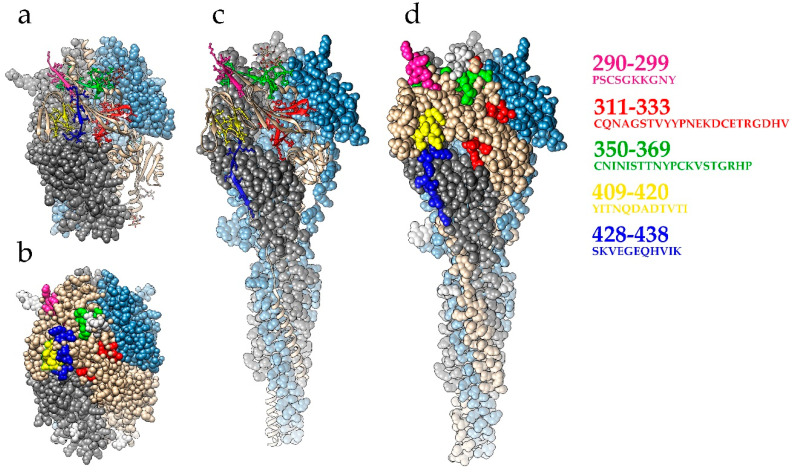
Mapping of B-cell epitopes predicted in [90] on the 3D structure of HMPV F protein. (**a**,**b**) F trimer, prefusion state (pdb ID 5wb0). Epitopes are shown as schematic structure elements (**a**) or in a surface presentation (**b**); (**c**,**d**) F trimer, postfusion state (pdb ID 5l1x). Epitopes are shown as schematic structure elements (**c**) or in a surface presentation (**d**). The fragment with epitope 88–102 (QLAREEQIENPRQSR) falls within a non-stable region and is not present in these structures. The figure was prepared using Chimera 1.14 software [101].

**Figure 4 pathogens-09-00481-f004:**
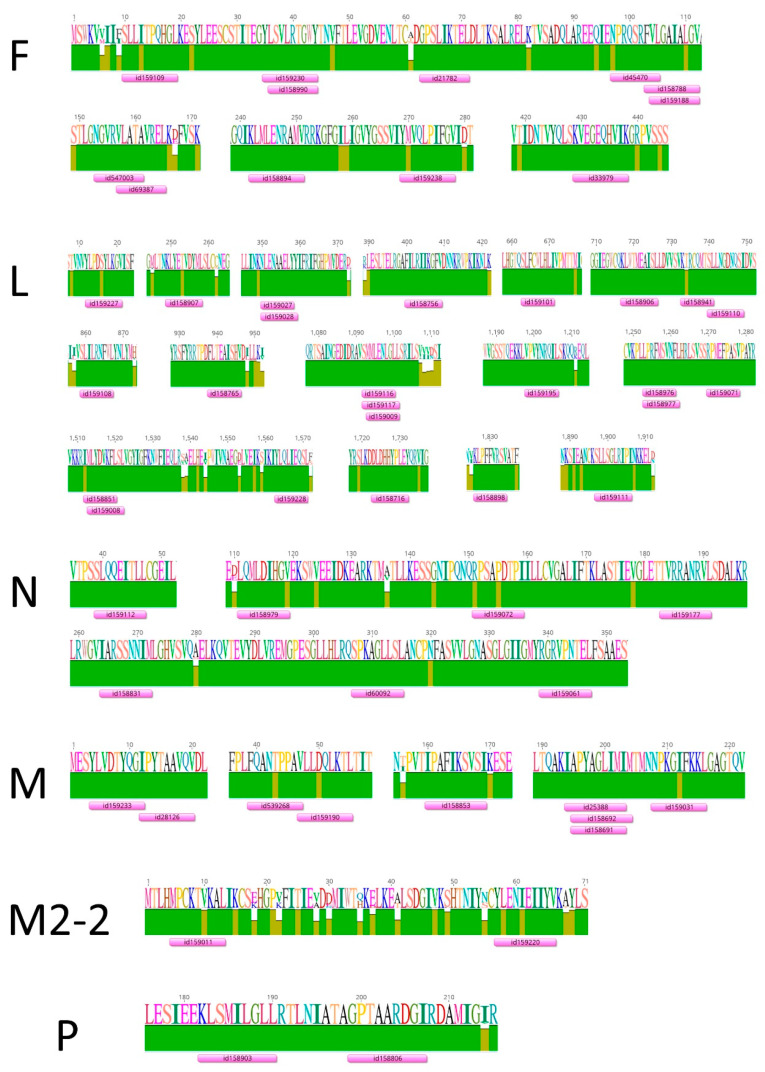
Localization of HMPV human CD8 T-cell epitopes in conserved regions of HMPV proteins. F, fusion protein; L, RNA-dependent RNA polymerase; N, nucleoprotein; M, matrix protein; M2-2, matrix protein 2-2; P, phosphoprotein. Full-genome sequences of 157 HMPV isolates belonging to each HMPV genotype were retrieved from the Virus Pathogen Database (ViPR, viprbrc.org). Human CD8 T-cell epitopes were selected from experimentally confirmed epitopes deposited at the Immune Epitope Database (iedb.org). Alignment of viral proteins, alignment of T-cell epitopes, and visualization were performed using Geneious 10.2.5 software.

**Table 1 pathogens-09-00481-t001:** B-cell neutralizing epitopes of human metapneumovirus deposited in the IEDB.

IEDB ID	Epitope Sequence	Reference
174346	L19, K20, E21, S22, Y23, L24, E25, E26, I31, E33, G34, P282, C283, W284, K312, T317, E345, R348, E349, N351, I352, N353, T356, N358, L378, T411, Q413, D414	[92,93]
101452	K386	[94]
101453	S132, G152	[94]
101454	V397	[94]
101455	A238, G239, I241, K242, L245	[94]
101456	A238, I241, K242, L245	[94]
101457	I177, N178, K179	[94]
101458	K242	[94]

**Table 2 pathogens-09-00481-t002:** List of conserved HMPV epitopes with confirmed binding to HLA class I molecules.

Protein (Position)	IEDB ID	Epitope Sequence	Allele	Method	Reference
F (10–19)	159109	SLLITPQHGL	HLA-A*02:01	MHC binding	[91]
F (35–44)	159230	YLSVLRTGWY	HLA-A*01:01	MHC binding	[91]
F (36–44)	158990	LSVLRTGWY	HLA-A*01:01	MHC binding	[91]
F (63–71)	21782	GPSLIKTEL	HLA-B*07:02	MHC binding	[91]
F (97–105)	45470	NPRQSRFVL	HLA-B*07:02	MHC binding	[91]
F (103–112)	158788	FVLGAIALGV	HLA-A*02:01	MHC binding	[91]
F (104–112)	159188	VLGAIALGV	HLA-A*02:01	MHC binding	[91]
F (153–161)	547003	NGVRVLATA	HLA-A*02:01	PBMC (51 chromium cytotoxicity)	[105]
F (157–165)	69387	VLATAVREL	HLA-A*02:01	PBMC (ELISPOT IFNγ release, 51 chromium cytotoxicity)	[96]
F (242–251)	158894	KLMLENRAMV	HLA-A*02:01	MHC binding	[91]
F (269–278)	159238	YMVQLPIFGV	HLA-A*02:01	MHC binding	[91]
F (429–438)	33979	KVEGEQHVIK	HLA class I	PBMC (ELISPOT IFNγ release, 51 chromium cytotoxicity)	[96]
L (12–21)	159227	YLPDSYLKGV	HLA-A*02:01	MHC binding	[91]
L (249–258)	158907	KLYETVDYML	HLA-A*02:01	MHC binding	[91]
L (350–358)	159027	NLENAAELY	HLA-A*01:01	MHC binding	[91]
L (350–359)	159028	NLENAAELYY	HLA-A*01:01	MHC binding	[91]
L (400–409)	158756	FILRIIKGFV	HLA-A*02:01	MHC binding	[91]
L (663–671)	159101	SLFCWLHLI	HLA-A*02:01	MHC binding	[91]
L (717–726)	158906	KLWTMEAISL	HLA-A*02:01	MHC binding	[91]
L (733–741)	158941	KTRCQMTSL	HLA-B*07:02	MHC binding	[91]
L (740–749)	159110	SLLNGDNQSI	HLA-A*02:01	MHC binding	[91]
L (859–867)	159108	SLILRNFWL	HLA-A*02:01	MHC binding	[91]
L (938–946)	158765	FLTEAISHV	HLA-A*02:01	MHC binding	[91]
L (1092–1100)	159116	SMMLENLGL	HLA-A*02:01	MHC binding	[91]
L (1092–1101)	159117	SMMLENLGLL	HLA-A*02:01	MHC binding	[91]
L (1093–1101)	159009	MMLENLGLL	HLA-A*02:01	MHC binding	[91]
L (1198–1206)	159195	VPVYNRQIL	HLA-B*07:02	MHC binding	[91]
L (1253–1261)	158976	LPRFMSVNF	HLA-B*07:02	MHC binding	[91]
L (1253–1262)	158977	LPRFMSVNFL	HLA-B*07:02	MHC binding	[91]
L (1270–1278)	159071	RPMEFPASV	HLA-B*07:02	MHC binding	[91]
L (1512–1520)	158851	IMLYDVKFL	HLA-A*02:01	MHC binding	[91]
L (1513–1522)	159008	MLYDVKFLSL	HLA-A*02:01	MHC binding	[91]
L (1563–1571)	159228	YLQLIEQSL	HLA-A*02:01	MHC binding	[91]
L (1723–1732)	158716	DLDHHYPLEY	HLA-A*01:01	MHC binding	[91]
L (1826–1834)	158898	KLPFFVRSV	HLA-A*02:01	MHC binding	[91]
L (1897–1906)	159111	SLLSGLRIPI	HLA-A*02:01	MHC binding	[91]
N (39–47)	159112	SLQQEITLL	HLA-A*02:01	PBMC/Tg mice (ELISPOT IFNγ release, qualitative binding)	[95]
N (111–119)	158979	LQMLDIHGV	HLA-A*02:01	MHC binding	[91]
N (151–159)	159072	RPSAPDTPI	HLA-B*07:02	MHC binding	[91]
N (183–191)	159177	TVRRANRVL	HLA-B*07:02	MHC binding	[91]
N (264–272)	158831	IARSSNNIM	HLA-B*07:02	MHC binding	[91]
N (307–315)	60092	SPKAGLLSL	HLA-B*07	PBMC (ELISPOT IFNγ release, 51 chromium cytotoxicity)	[96]
N (339–347)	159061	RGRVPNTEL	HLA-B*07:02	MHC binding	[91]
M (4–12)	159233	YLVDTYQGI	HLA-A*02:01	MHC binding	[91]
M (12–20)	28126	IPYTAAVQV	HLA-B*07	PBMC (ELISPOT IFNγ, 51 chromium cytotoxicity)	[96]
M (39–47)	539268	FQANTPPAV	HLA-A*02:01	PBMC/Tg mice (ELISPOT IFNγ release, qualitative binding, pathogen burden after challenge)	[95]
M (47–55)	159190	VLLDQLKTL	HLA-A*02:01	MHC binding	[91]
M (160–169)	158853	IPAFIKSVSI	HLA-B*07:02	MHC binding	[91]
M (194–203)	25388	IAPYAGLIMI	HLA class I	PBMC (ELISPOT IFNγ release, 51 chromium cytotoxicity)	[96]
M (195–203)	158691	APYAGLIMI	HLA-B*07:02	Tg mice (ICS IFNγ, pathogen burden after challenge, degranulation)	[46,47]
M (195–204)	158692	APYAGLIMIM	HLA-B*07:02	MHC binding	[91]
M (208–216)	159031	NPKGIFKKL	HLA-B*07:02	MHC binding	[91]
M2-2 (5–13)	159011	MPCKTVKAL	HLA-B*07:02	MHC binding	[91]
M2-2 (57–66)	159220	YLENIEIIYV	HLA-A*02:01	MHC binding	[91]
P (182–190)	158903	KLSMILGLL	HLA-A*02:01	MHC binding	[91]
P (199–207)	158806	GPTAARDGI	HLA-B*07:02	MHC binding	[91]

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
