# Peer review of "Prospects of and Barriers to the Development of Epitope-Based Vaccines against Human Metapneumovirus"

_pathogens, 2020, doi:10.3390/pathogens9060481_

Round 1

Reviewer 1 Report

In this comprehensive review, the current state of our progress toward the development of epitope-based vaccines against human metapneumovirus is discussed.  Attention is given to both antibody-binding and T cell-specific epitopes.  The review is considered highly comprehensive in both its enumeration and illustration of the known B cell epitopes and establishment of the need for the further identification of additional T cell-specific epitopes.  The point is made that the importance of the latter is especially driven by the knowledge of the ability of HMPV to modulate the immune response, in particular its ability to impair the CD8 CTL response and, thus, the ability to generate a long-lasting immunological memory.  In my opinion, the review presents a highly detailed analysis of the topic and presents the status of the current state of HMPV vaccinology in a clear and through fashion. 

Author Response

In this comprehensive review, the current state of our progress toward the development of epitope-based vaccines against human metapneumovirus is discussed.  Attention is given to both antibody-binding and T cell-specific epitopes.  The review is considered highly comprehensive in both its enumeration and illustration of the known B cell epitopes and establishment of the need for the further identification of additional T cell-specific epitopes.  The point is made that the importance of the latter is especially driven by the knowledge of the ability of HMPV to modulate the immune response, in particular its ability to impair the CD8 CTL response and, thus, the ability to generate a long-lasting immunological memory.  In my opinion, the review presents a highly detailed analysis of the topic and presents the status of the current state of HMPV vaccinology in a clear and through fashion. 

Authors’ response: The authors are very thankful to the Reviewer for the positive feedback.

Reviewer 2 Report

“Prospects of and Barriers to the Development of Epitope-Based Vaccines against Human Metapneumovirus’

This manuscript provides a detailed analysis of the available designs of hMPV vaccines using vector delivery systems. Human metapneumovirus and respiratory syncytial virus are two major causes of lower respiratory tract infection in the pediatric population. hMPV also responsible for the mortality in immunocompromised patients, thus has high relevance in developing a vaccine for hMPV. Despite the efforts of researchers, still no vaccine or specific prophylaxis for hMPV is available for human use.

This is a well written review that has covered all possible scenarios of the vaccine development for hMPV, the authors have emphasized the possible target regions and the development of a future vaccine towards the human CD4 T-cell epitopes of hMPV.

  1. Introduction add: Human metapneumovirus (hMPV), a negative-sense single-stranded RNA virus, belongs to the Paramyxoviridae family that includes respiratory syncytial virus (RSV) and parainfluenza. Lines 108-118: may be moved to introduction
  2. Lines 92-94: rewrite the sentence.
  3. Line 459: higher neutralizing antibody titers (Tfh) and the enhancement of the CTL response (Th1). Is that Th2?

Author Response

“Prospects of and Barriers to the Development of Epitope-Based Vaccines against Human Metapneumovirus’

This manuscript provides a detailed analysis of the available designs of hMPV vaccines using vector delivery systems. Human metapneumovirus and respiratory syncytial virus are two major causes of lower respiratory tract infection in the pediatric population. hMPV also responsible for the mortality in immunocompromised patients, thus has high relevance in developing a vaccine for hMPV. Despite the efforts of researchers, still no vaccine or specific prophylaxis for hMPV is available for human use.

This is a well written review that has covered all possible scenarios of the vaccine development for hMPV, the authors have emphasized the possible target regions and the development of a future vaccine towards the human CD4 T-cell epitopes of hMPV.

Authors’ response: We strongly appreciate the comments and suggestions given by the Reviewer and have adjusted the manuscript accordingly.

Introduction add: Human metapneumovirus (hMPV), a negative-sense single-stranded RNA virus, belongs to the Paramyxoviridae family that includes respiratory syncytial virus (RSV) and parainfluenza.

Authors’ response: Corrected together with the next remark (lines 108-118 were moved to introduction, this text block contains information regarding systematics of Metapneumovirus)

Lines 108-118: may be moved to introduction

Authors’ response: Corrected, lines 108-118 were moved to appropriate introduction section.

Lines 92-94: rewrite the sentence.

Authors’ response: The sentence was corrected.

Line 459: higher neutralizing antibody titers (Tfh) and the enhancement of the CTL response (Th1). Is that Th2?

Authors’ response: This abbreviation was corrected in the text to avoid misunderstanding. In this case, the abbreviation «Tfh» was used to denote follicular helper T-cells, a Bcl6-dependent differentiated subpopulation of CD4+ T lymphocytes, interacting with B-cells in germinal centers.

Reviewer 3 Report

Overall the review is very well-written and thorough.  I have attached a pdf file with a few comments about wording and a couple ideas that are not fully explained throughout the manuscript.  The only large gap appears at the very end of the manuscript in the conclusions where the ideas seem to end without providing a sentence or two for closure of the ideas.  Adding 1-2 sentences about the progress of HMPV vaccines would provide an ending for the reader.

Author Response

Overall the review is very well-written and thorough.  I have attached a pdf file with a few comments about wording and a couple ideas that are not fully explained throughout the manuscript.  The only large gap appears at the very end of the manuscript in the conclusions where the ideas seem to end without providing a sentence or two for closure of the ideas.  Adding 1-2 sentences about the progress of HMPV vaccines would provide an ending for the reader.

Authors’ response: We are very thankful to the Reviewer for the careful analysis of our paper and providing valuable comments. We strongly appreciate the comments and suggestions given by Reviewer and have adjusted the manuscript accordingly. The proposed changes were made. The heading of 3.3 section was changed in accordance to reviewer comment. Typos were corrected as well.

Line 290:  It is unclear whether the authors believe whether attenuated influenza viruses could be used for delivery of HMPV epitopes.  Another sentence clarifying this idea would be helpful to assist the reader in understanding the potential utility of this vaccination strategy.

Authors’ response: Two sentences explaining the idea of perspectives of usage influenza vector for HMPV vaccine development were added: Like HMPV, influenza virus is a respiratory pathogen and can induce local immunity barrier in the airways, making it a promising platform for the delivery of HMPV epitopes to the site of infection. In addition, attenuated influenza viruses are known to be effective inducers of cross-reactive T-cell responses [82,83].

Line 450: Is there a specific reason that the authors are recommending a virus-vectored vaccine over the other approaches that were identified?  While the virus-vectored vaccines are promising, an explanation for the reasoning would be helpful for the reader.

Authors’ response: We strongly appreciate to Referee for this remark. The following sentence was added to address the issue raised by the reviewer: Viral vectors are a promising platform for the development of vaccines against other viruses due to effective immunogen delivery and stimulation of the appropriate components of immune system specific for intracellular pathogens.

Line 471: This conclusion feels somewhat rushed.  Another 1-2 sentences about the overall direction of HMPV vaccine research would provide some closure to the manuscript. 

Authors’ response: We are grateful to Reviewer for this remark. The Conclusions section was expanded with summarizing sentences: To date, the most promising approaches to the development of HMPV vaccines look like the use of live antigen delivery systems (vectors or live attenuated viruses) with an accurate immunogen design to ensure effective T-cell memory formation. Additional studies of immunodominant CD8 and CD4 epitopes in HMPV proteins are needed; the role of CD4 T-cell subpopulations has to be estimated in details. Several methods have been described to evaluate HMPV T-cell memory in humans, as well as the efficacy of T-cell vaccines in experimental animal models. These approaches can help develop effective epitope-based vector vaccines for the prevention of HMPV.